# IMPROVING SOURCE EXTRACTION WITH DIFFUSION AND CONSISTENCY MODELS

## ABSTRACT

In this work, we demonstrate the integration of a score-matching diffusion model into a deterministic architecture for time-domain musical source extraction, resulting in enhanced audio quality. To address the typically slow iterative sampling process of diffusion models, we apply consistency distillation and reduce the sampling process to a single step, achieving performance comparable to that of diffusion models, and with two or more steps, even surpassing them. Trained on the Slakh2100 dataset for four instruments (bass, drums, guitar, and piano), our model shows significant improvements across objective metrics compared to baseline methods. Sound examples are available at https://consistency-separation.github.io/.

## 1 INTRODUCTION

An audio source extraction refers to the process of isolating or extracting a specific sound source from a mixture of audio signals. The source extraction task is closely related to source separation; while extraction focuses on isolating a specific target source, source separation involves separating all components. These techniques are crucial in various fields, including speech separation, noise reduction, music processing, music analysis, transcription, and more. Different from other fields in audio, music is characterized by a strong interdependence between the sources (stems) in a mixture. Stems interact not only in terms of timing and rhythm but also through their spectral and harmonic content. In addition, masking effects frequently occur in audio mixing (Moore, 2012), where louder sounds can obscure quieter ones, and overlapping frequencies—particularly in low-frequency instruments like bass and kick drum—can interfere with each other, reducing perceptual clarity. These challenges often make isolating individual sounds particularly difficult. However, humans have a natural ability to focus on specific sound sources, as illustrated by the "cocktail party effect" in speech (Cherry, 1953), and similarly in music, where listeners can focus on specific instruments (Bregman, 1984; McAdams & Bregman, 1979), discerning what each one is playing. The latter is especially well-developed in professional musicians and composers, who often extract, transcribe, or analyze individual musical elements as part of their creative process.

Recent advances in deep learning technology have significantly impacted the field of audio source extraction and separation, leading to substantial improvements in its quality. There are two primary approaches to the field using machine learning. The first approach involves deterministic discriminative models (Choi et al., 2021; Défossez, 2021; Lluís et al., 2019; Gusó et al., 2022; Luo & Mesgarani, 2019; Défossez et al., 2019; Takahashi et al., 2018), which typically use mixtures for conditioning in the training and inference process and learn how to derive one or more sources from mixtures. On the other hand, generative models (Zhu et al., 2022; Subakan & Smaragdis, 2018; Kong et al., 2019; Narayanaswamy et al., 2020; Jayaram & Thickstun, 2021; Postolache et al., 2023a; Manilow et al., 2022; Postolache et al., 2023b; Kavalerov et al., 2019; Wisdom et al., 2020) generally learn a prior distribution of sources and use the mixture during inference to generate separate sources. Recently, there have been several attempts to apply diffusion models (Sohl-Dickstein et al., 2015; Ho et al., 2020) to audio source separation and extraction tasks (Scheibler et al., 2023; Lutati et al., 2023; Huang et al., 2023; Yu et al., 2023; Hirano et al., 2023; Plaja-Roglans et al., 2022; Mariani et al., 2024).

We propose an extension of the deterministic mixture-conditional musical source extraction model by incorporating a generative methods. First, we train a deterministic model. However, deterministic models relying solely on the mixture to regressively derive the sources often struggle with imperfect

separation and source leakage, caused by the complex relationships between musical sound sources and masking effects within the mixture. This creates an upper performance limit for deterministic models. To address this limitation, we introduce a generative element to our system: a denoising score-matching diffusion model (Song & Ermon, 2019) that enhances the extracted sources, providing a generative final touch. Since generative models can synthesize data from scratch, we hypothesize that incorporating a generative component would help model to reconstruct missing information, reduce source leakage and farther improve the quality of source extraction. The introduction of diffusion models into our system required an iterative sampling procedure, which increased the inference time for the overall model. To speed up the generation process, we adapted methodologies presented in Consistency Models (CM) (Song et al., 2023) and Consistency Trajectory Models (CTM) (Kim et al., 2024), applying Consistency Distillation (CD) to our diffusion model, thereby reducing the number of denoising steps.

We trained our model on the Slakh2100 (Manilow et al., 2019) dataset, focusing on four instruments: bass, drums, guitar, and piano. In our experiments, we compared our model with baseline models Demucs (Défossez et al., 2019), Demucs+Gibbs (Manilow et al., 2022), ans MSDM (Mariani et al., 2024) and demonstrated significant improvements in the objective metrics of music separation. Our diffusion model showed enhanced extraction quality over the deterministic model. Furthermore, our CD model achieved accelerated single-step denoising without any loss of quality. The consistency model also provided flexibility in trading off between quality and speed, delivering improved results over the diffusion model with only 2-4 steps and a minimal increase in time compared to the deterministic model. Notably, our CD model outperformed the diffusion model without the use of adversarial training, which contrasts with CTM, where superior performance in the image domain was achieved by incorporating the GAN (Goodfellow et al., 2014) component into the training procedure along with the denoising loss.

**Contributions** Our work presents three main contributions: **(i)** We bridge the gap between deterministic and generative source extraction methods, showing they can complement each other and improve overall model performance. **(ii)** To the best of our knowledge, we are the first to introduce a consistency framework to the raw audio domain and demonstrate a student CD model outperforming the teacher diffusion model without the use of additional GAN loss. **(iii)** We achieve significant improvements in musical source extraction and set a new benchmark on the Slakh2100 dataset.

## 2 RELATED WORK

Audio source extraction and separation fields have seen significant advances, driven by the development of deep learning techniques and the availability of large-scale datasets. A significant body of work in the field focuses on time-frequency representations, such as the Short-Time Fourier Transform (STFT). Initially, STFT models concentrated on speech extraction (Grais et al., 2014), later expanding to include music (Uhlich et al., 2015; 2017; Liu & Yang, 2018; Takahashi & Mitsufuji, 2017; Nugraha et al., 2016). State-of-the-art models for music source separation in the spectrogram domain include MMDenseLSTM (Takahashi et al., 2018), which combines convolutional and recurrent networks, D3Net (Takahashi & Mitsufuji, 2020), which uses dilated convolutions, and Spleeter (Hennequin et al., 2020), based on a U-Net architecture.

In recent years, there has been a shift in focus towards models that operate directly on raw waveforms. The pioneering waveform-based audio separation model from Lluís et al. (2019), built on the WaveNet architecture (van den Oord et al., 2016), was followed by Wave-U-Net (Stoller et al., 2018), based on the U-Net architecture, with both models underperforming compared to spectrogram-based models at the time. Conv-TasNet (Luo & Mesgarani, 2019), utilizing stacked dilated 1-D convolutional blocks, was the first waveform-based model to surpass spectrogram-based approaches in speech source separation. Demucs (Défossez et al., 2019) extended the U-Net architecture by incorporating bidirectional LSTM layers between the encoder and decoder, surpassing all existing state-of-the-art architectures at the time. Building on Demucs, the source separation problem was reframed as an Orderless Neural Autoregressive Density Estimator (NADE) by Demucs+Gibbs (Manilow et al., 2022), using a Gibbs sampling procedure to iteratively improve separation performance by conditioning each source estimate on those from previous steps. These latest two works serve as the primary baselines for our approach.

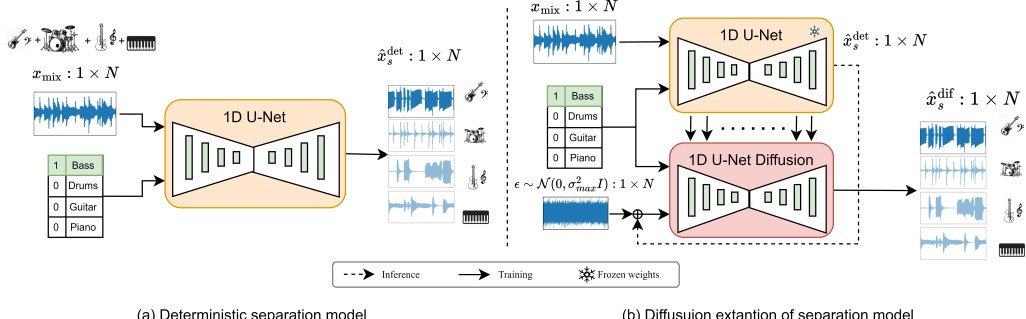

(a) Deterministic separation model       (b) Diffusuion extantion of separation model

Figure 1: **Diagram illustrating our proposed method.** (a) First, we train a mixture-conditional deterministic source extraction model. (b) Next, we introduce a denoising score-matching diffusion model, conditioned both on the features extracted by the deterministic model and instrument label, which farther enhances extracted audio quality through noise addition and removal.

Similar to our work, MSDM (Mariani et al., 2024) introduced a denoising score-matching (Song & Ermon, 2019) diffusion-based U-Net architecture for the musical sourse separation in the waveform domain. MSDM operates in a multichannel manner and is fully generative, proposing a novel inference-time conditioning scheme based on the Dirac delta function for posterior sampling. Unlike our approach and most separation models, MSDM is also capable of synthesizing music and creating arrangements, which makes it not a direct comparison to our work. Nevertheless, we include it in our baselines due to the architectural and methodological similarities.

## 3 METHOD

Let $x_{\text{mix}}$ represent a time-domain audio mixture containing $S$ individual tracks $x_s \in 1 \times N$, where $N$ is the number of audio samples and $s \in \{1, \ldots, S\}$ identifies each source. The mixture is defined as $x_{\text{mix}} = \sum_{s=1}^{S} x_s$. The source extraction problem is to minimize some loss function $\mathcal{L}$ that measures the average error between the true source $x_s$ and its corresponding prediction $\hat{x}_s$. Having all $S$ sources extracted, the task becomes one of source separation.

### 3.1 DETERMINISTIC MODEL

We develop and train a deterministic source extraction model $f_\theta$ using a U-Net encoder-decoder architecture, as shown in the left side of Fig. 1. The U-Net, consisting of 1D convolutional layers with skip connections, is a common choice for time-domain source separation (Stoller et al., 2018; Défossez et al., 2019; Manilow et al., 2022). Our model has two inputs: one for the mixture signal $x_{\text{mix}}$ and another for the instrument of interest $s$. We train our deterministic model $f_\theta$ with the following loss function:

$$\mathcal{L}(\theta) = \mathbb{E}_{s, x_{\text{mix}}} \|x_s - \hat{x}_s^{\text{det}}\|_2^2, \tag{1}$$

where $\hat{x}_s^{\text{det}} = f_\theta(x_{\text{mix}}, s)$ is the prediction for source $s$.

### 3.2 DIFFUSION MODEL

After training the deterministic model $f_\theta$, we freeze its parameters and integrate it into a larger system that includes a score-matching diffusion model $g_\phi$, as depicted in right side of Fig. 1. The diffusion model $g_\phi$ has the same architecture as the deterministic model but takes four inputs instead of two: (1) a noisy version of $x_s$, (2) the target label $s$, (3) the scale of noise $\sigma$ added to the $x_s$, and (4) the intermediate features $\bar{x}_s^{\text{det}}$ extracted by the frozen deterministic model. We train the diffusion model $g_\phi$ with the Denoising Score Matching (DSM) loss:

$$\mathcal{L}_{\text{DSM}}(\phi) = \mathbb{E}_{s, x_{\text{mix}}, t} \|x_s - g_\phi(x_s + \sigma_t \epsilon, s, \sigma_t, \bar{x}_s^{\text{det}})\|_2^2, \tag{2}$$

where $\epsilon \sim \mathcal{N}(0, I)$ is noise sampled from a standard Gaussian distribution, and $\sigma_t := \sigma(t)$ is a monotonically increasing function that defines the noise step and the scale of the added noise.

The inference of $g_\phi$ is an iterative process of solving a numerical Ordinary Differential Equation (ODE) over $T$ steps. At each step, the diffusion model $g_\phi$ polishes $\hat{x}_s^{\text{det}}$ using intermediate features $\bar{x}_s^{\text{det}}$ and appropriate amount of noise $\sigma_t \epsilon$ for $\sigma : [1, T] \rightarrow [\sigma_{\min}, \sigma_{\max}]$ where $\sigma_{\min}$ and $\sigma_{\max}$ show minimum and maximum noise levels, respectively. The iterative process begins with the maximum noise level $\sigma_{\max}$, setting the initial prediction $\hat{x}_{s,T-1}^{\text{dif}} = \texttt{Solver}_1(\hat{x}_s^{\text{det}} + \sigma_{\max}\epsilon, s, \sigma_T, \bar{x}_s^{\text{det}}; g_\phi)$ and continues with the following update rule:

$$\hat{x}_{s,t-1}^{\text{dif}} = \texttt{Solver}_1(\hat{x}_{s,t}^{\text{dif}}, s, \sigma_t, \bar{x}_s^{\text{det}}; g_\phi), \tag{3}$$

where $\texttt{Solver}_k(\ldots; g_\phi)$ denotes $k$ steps of any ODE solver that uses $g_\phi$ score-based model for data denoising. This process continuous until reaching clean $\hat{x}_{s,0}^{\text{dif}}$.

For more details on score-based diffusion models, please see Appendix A.1.

### 3.3 Consistency Model

To mitigate the latency introduced by the diffusion model in the inference process and make 1-2 steps generation possible, we adopt CD (Song et al., 2023; Kim et al., 2024). In this approach, our consistency model $g_\omega$ is designed as an exact replica of the diffusion model and is trained using a pretrained diffusion model $g_\phi$ as a teacher. Requiring inference of diffusion teacher model, CD is a designed as discrete process with $t \in [1, T]$, where $T$ denotes a total number of steps. We iteratively apply an ODE solver to predict progressively less noisy samples along the trajectory:

$$\hat{x}_{s,t-h}^{\text{dif}} = \texttt{Solver}_h(x_s + \sigma_t \epsilon, s, \sigma_t, \bar{x}_s^{\text{det}}; g_\phi), \tag{4}$$

where $h \in [1, t]$ is the number of ODE steps used in distilation process. This prediction is then used to calculate the target in our CD optimization, with the following loss:

$$\mathcal{L}_{\text{CD}}(\omega) = \mathbb{E}_{t,h} \| \underbrace{g_{\text{sg}(\omega)}(\hat{x}_{s,t-h}^{\text{dif}}, s, \sigma_{t-h}, \bar{x}_s^{\text{det}})}_{target} - \underbrace{g_\omega(x_s + \epsilon\sigma_t, s, \sigma_t, \bar{x}_s^{\text{det}})}_{prediction} \|_2^2, \tag{5}$$

where $\texttt{sg}(\omega)$ denotes the stop-gradient running EMA (Exponential Moving Average) of $\omega$ during optimization, updated as $\texttt{sg}(\omega) \leftarrow \texttt{stopgrad}(\mu\texttt{sg}(\omega) + (1-\mu)\omega)$, with $\mu$ denoting update rate.

Inspired by the success of introducing direct signals from data by use of the DSM loss in CTM (Kim et al., 2024), we adopted similar idea for our model and applied the DSM loss from equation 2. Our final loss is formulated as:

$$\mathcal{L}(\omega) = \mathcal{L}_{\text{CD}}(\omega) + \lambda_{\text{DSM}}\mathcal{L}_{\text{DSM}}(\omega), \tag{6}$$

where $\lambda_{\text{DSM}}$ is a balancing term between two losses.

A more detailed account to CD techniques is provided in Appendix A.2.

## 4 Experimental Setup

### 4.1 Dataset

For our training and evaluation experiments, we used the Slakh2100 dataset (Manilow et al., 2019), a widely recognized and extensively used benchmark for music source separation. Our choice of dataset was driven by the high data requirements of diffusion models, making Slakh an ideal candidate for training. Slakh consists of 2100 tracks, with 1500 allocated for training, 375 for validation, and 225 for testing. It is a synthetically generated multi-track audio waveform dataset created from MIDI files using virtual instruments. The dataset contains 31 instrument classes. We downsampled the audio to 22kHz and used a duration of $N = 262144$ samples, which approximately corresponds to 11.9 seconds of audio file length. We focused on four instruments—Bass, Drums, Guitar, and Piano—due to their prevalence in the dataset and to ensure direct comparability with baseline models. The presence of these instruments in the dataset is 94.7% for Bass, 99.3% for Drums, 100.0% for Guitar, and 99.3% for Piano.

In addition to Slakh2100, we also trained and evaluated our models using the MUSDB18 (Rafii et al., 2017) dataset, which is another widely used data for music source separation. The dataset consists of 150 tracks, with 100 designated for training and 50 for testing, totaling approximately 10 hours of professional-grade audio. Each track is divided into stems: Bass, Drums, Vocals, and Other. For the results on MUSDB18 dataset please refer to Appendix C.

Table 1: **SI-SDR$_I$ (dB — higher is better) results for source separation on the Slakh2100 test set.** We compare our deterministic model, diffusion, and consistency distillation results with our baselines for all stem categories. 'All' reports the average across the four stems.

| Model | Bass | Drums | Guitar | Piano | All |
|---|---|---|---|---|---|
| Demucs (Défossez et al., 2019; Manilow et al., 2022) | 15.77 | 19.44 | 15.30 | 13.92 | 16.11 |
| Demucs + Gibbs (512 steps) (Manilow et al., 2022) | 17.16 | 19.61 | 17.82 | 16.32 | 17.73 |
| ISDM (Dirac with correction)(Mariani et al., 2024) | 19.36 | 20.90 | 14.70 | 14.13 | 17.27 |
| MSDM (Dirac with correction)(Mariani et al., 2024) | 17.12 | 18.68 | 15.38 | 14.73 | 16.48 |
| Deterministic model | 20.04 | 20.88 | 23.82 | 20.82 | 21.39 |
| Deterministic model double | 20.93 | 20.97 | 23.56 | 21.14 | 21.65 |
| Diffusion ($T = 5, \sigma_{\max} = 0.01, R = 2$) | 21.06 | 21.77 | 25.35 | 22.17 | 22.58 |
| CD onestep ($T = 1, \sigma_{\max} = 0.01244$) | 20.99 | 21.91 | 26.10 | 22.73 | 22.93 |
| CD multistep ($T = 2, \sigma_{\max} = 0.2497$) | 21.97 | 22.19 | 27.34 | 23.24 | 23.68 |
| CD multistep ($T = 4, \sigma_{\max} = 0.2495$) | **22.39** | **22.45** | **28.09** | **23.96** | **24.22** |

## 4.2 Model Architecture and Training Overview

All of our models—Deterministic, Diffusion, and Consistency—are based on a U-Net backbone. Our U-Net design follows the structure of generative U-Net models from Moûsai (Schneider et al., 2024) and MSDM (Mariani et al., 2024), with modifications.

The U-Net for the deterministic model was modified to accept a one-hot encoded instrument label as a conditioning input, with a size of $S = 4$. Our hyperparameter settings for the deterministic model mostly follow those used in MSDM, with the exception of using a single channel instead of four. We trained the deterministic model using the Adam optimizer with a learning rate of $1 \times 10^{-4}$ for 170 epochs, until convergence.

For the diffusion model, we used the same backbone U-Net architecture as in the deterministic model, wrapping it within the diffusion model framework. We added additional conditioning by incorporating the deterministic model as a feature extractor, integrating the extracted features into the diffusion model layer by layer. The preconditioning and training procedures for the diffusion model largely followed the method introduced in the EDM framework (Karras et al., 2022). We trained the diffusion model using the Adam optimizer with a learning rate of $1 \times 10^{-4}$ for 280 epochs until convergence.

The consistency model is an exact replica of the diffusion model and, for effective distillation, is initialized using the pre-trained weights of the diffusion model. For the Consistency model, we applied consistency distillation as described in the original work (Song et al., 2023), but with some modifications. Specifically, we did not use schedule functions for $T$ or $\mu$, the total number of steps, and the update rate of the EMA for the stop-gradient student model, and instead kept these values fixed at $T = 18$ and $\mu = 0.999$ throughout the experiments. We increased the number of ODE steps to $h \leq 17$, compared to $h = 1$ in the original. We incorporated settings from the CTM framework (Kim et al., 2024) to balance the losses in equation 6. The consistency model was trained using the Rectified Adam optimizer (Liu et al., 2020), without applying learning rate decay, warm-up, or weight decay, with a fixed learning rate of $1 \times 10^{-5}$ for 50 epochs.

For more detailed technical account of our models, please first refer to Appendix A, which introduces preliminary foundational concepts, notations, algorithms, and mathematical formulations of score-based models and consistency models, and then refer to Appendix B for more detailed implementation, hyperparameters, training, and inference settings.

## 5 Results

### 5.1 Model Performance Comparison

We compared our deterministic model, diffusion model, and CD results on Slakh2100 with the baselines Demucs (Défossez et al., 2019), Demucs + Gibbs (Manilow et al., 2022), and two different

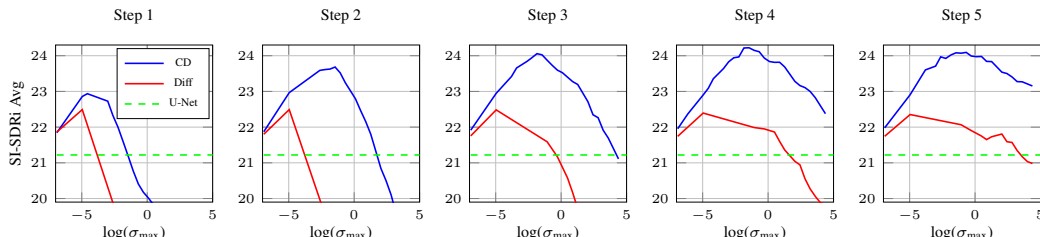

Figure 2: **SI-SDRi Avg. vs Log($\sigma_{\text{max}}$) for CD and Diffusion Models across 5 Steps**. Each subplot compares the performance of the diffusion model (red) and the consistency distillation model (blue) across different numbers of denoising steps, with a dashed line representing the performance of the deterministic model. The x-axis represents $\sigma_{\text{max}}$, the starting noise levels for the models, given in a logarithmic scale.

methods from MSDM (Mariani et al., 2024), including supervised and weakly-supervised models with the Dirac algorithm. For direct comparison, we use the scale-invariant SDR improvement (SI-SDR$_I$) metric (Roux et al., 2019) and adopted the exact same evaluation procedure as implemented in MSDM and Demucs + Gibbs. We evaluate over the test set of Slakh2100, using a sliding window with chunks of 4 seconds in length and a 2-second overlap. Additionally, we filter out silent chunks and chunks consisting of only one source, due to the poor performance of SI-SDR$_I$ on such segments.

Results are reported in Table 1 and show the following:

*(i)* Our vanilla deterministic model outperforms the baselines. We attribute this performance to the slightly larger size and improved architecture of our model (using self-attention instead of LSTM, as in Demucs) compared to the baseline models.

*(ii)* Adding a diffusion model to the system further improved separation quality. We hypothesized that incorporating a generative model would help reconstruct missing information, reduce leakage and further improve the quality of source extraction. Indeed, we found that our system, with the addition of a score-based diffusion generative model, outperformed deterministic model, improving the system's overall performance quality by approximately 1.2 dB. The diffusion model achieved the best performance using a stochastic sampler from EDM (Karras et al., 2022) for inference, with $T = 5$ timesteps, $R = 2$ correction steps (resulting in 10 actual steps), and a starting noise standard deviation of $\sigma_{\text{max}} = 0.01$. For a fair comparison, we also evaluated a double deterministic model without diffusion to assess whether the improvement was due to increased model size or diffusion itself. As shown in the 5th row of the table, the double deterministic model offered only a slight, almost negligible, improvement over the single one. The reported number is the observed best performance, around 150 epochs, after which a clear decline in quality accrued, suggested overfitting. This suggests that the improvement with diffusion models is not merely due to increased model size.

*(iii)* Our CD model not only successfully accelerates performance but also further improves quality of source extraction. As observed in the last three rows of the table, CD with 1 step maintains the best performance of the diffusion model. Furthermore, CD with 2 steps adds almost 1 dB improvement compared to the diffusion, demonstrating the "student beating the teacher" effect, which we attribute to the use of the DSM loss. To the best of our knowledge, we are the first to demonstrate a CD model outperforming diffusion without the use of GANs, wich was demostrated for image data in CTM(Kim et al., 2024). Finally, the last row shows the best-performing CD model with $T = 4$ denoising steps, which provides an additional improvement over the 2-step CD, leading to a dramatic 3 dB improvement compared to deterministic model and 6.5 dB improvement compared to the best baseline model, setting a new benchmark for music source separation on the Slakh2100 dataset.

### 5.2 DIFFUSION VS CD

We conducted a hyperparameter search to evaluate the efficiency of CD model compared the diffusion at a lower number of steps. We performed sweeps with number of steps $T \in [1, 5]$ (Note: for the diffusion model, the solver has correction steps and the number of actual denoising steps $T \times R$ is used) and starting noise level $\sigma_{\text{max}} \in [0.001, 80.0]$ for both models. Figure 2 presents the average results of SI-SDRi over all instrument classes for both models across these parameters, showing $\sigma_{\text{max}}$ on log scale. Although the diffusion model shows some improvement over the deterministic model

Table 2: **Performance and efficiency comparison.** We compare the inference times for separating a single ~12s mixture, number of parameters, and real-time factors (RTF) of our models with the baselines. The inference times of our models are multiplied by 4 as our model performs source extraction, requiring 4 iterations to generate all tracks from the mixture.

| Model | Inference Time (s) | # of parameters | RTF |
|---|---|---|---|
| Demucs (no shift trick) (Défossez et al., 2019) | 0.1285 | 265.7M | 0.010 |
| Demucs + Gibbs (512 steps) (Manilow et al., 2022) | $0.1285 \times 512 = 65.80$ | $\sim 265.7$M | 5.529 |
| Demucs + Gibbs (256 steps) (Manilow et al., 2022) | $0.1285 \times 256 = 32.9$ | $\sim 265.7$M | 2.764 |
| ISDM (correction) (Mariani et al., 2024) | $4.6 \times 4 = 18.4$ | 405M $\times 4$ | 1.546 |
| MSDM (correction) (Mariani et al., 2024) | 4.6 | 405M | 0.386 |
| Deterministic Model | $0.0285 \times 4 = 0.114$ | 405M | 0.009 |
| Deterministic Model double | $0.0570 \times 4 = 0.228$ | 405M $\times 2$ | 0.019 |
| Diffusion $N \times R = 10$ steps | $0.3498 \times 4 = 1.399$ | 405M $\times 2$ | 0.117 |
| CD $T = 1$ steps | $0.0690 \times 4 = 0.276$ | 405M $\times 2$ | 0.023 |
| CD $T = 2$ steps | $0.0963 \times 4 = 0.385$ | 405M $\times 2$ | 0.032 |
| CD $T = 4$ steps | $0.1547 \times 4 = 0.618$ | 405M $\times 2$ | 0.051 |

benchmark (green dashed line at 21.22 dB), it does not perform optimally at higher noise levels and low step counts, particularly in the 1-2 step scenarios, peaking only at $T = 10$ actual steps (as reported in Table 1). In contrast, the CD model not only achieves performance equal to the diffusion model's best with just one step, but also consistently outperforms the diffusion model across all $\sigma_{\max}$ values and steps, providing further evidence of the student surpassing the teacher in our experiments.

## 5.3 SPEED AND MODEL EFFICIENCY

We compared our models with baselines in terms of speed and efficiency. Table 2 shows inference time, and real-time factor (RTF) for separating a 11.9-second audio mixture. Unlike Demucs, Demucs+Gibbs, and MSDM, which are separation models and extract all 4 tracks simultaneously, our models extracts one source at a time, thus we multiply inference times by 4. We additionally report the number of parameters of the models for the efficiency comparison.

Despite need for four iterations, our deterministic model shows the best speed, with an inference time of 0.114 seconds and an RTF of 0.009, comparable in performance with Demucs. This is expected, as Demucs and our deterministic model both are simple discriminative models with relatively similar model sizes. It is worth noting that, in the table, we report Demucs without the shift trick, which is the fastest but not the best-performing configuration for Demucs. The shift trick, as described in Défossez et al. (2019), involves running inference multiple times with different input shifts, which slows down the process by a factor proportional to the number of shifts used. In practice, this can potentially make inference 2 to 10 times slower, highlighting the advantageous edge of our deterministic model.

Our diffusion model, while slower than our deterministic model with 1.399 seconds and an RTF of 0.117, still outperforms generative baselines like ISDM and MSDM, which exhibit significantly longer inference times, the ISDM model, in particular, being both parameter-heavy and time-consuming. Another semi-generative method, Demucs+Gibbs, requiring many inference steps, also shows significantly slower performance than our diffusion models.

Our CD models, especially at $T = 1$, achieve an RTF of 0.023, only twice as slow as the deterministic models and comparable with double deterministic model, while providing significant audio quality improvement. Notably, our best-performing CD model with 4 steps, though 5 times slower than the deterministic model, remains still significantly faster than other diffusion-based models (including ours) and far below the real-time generation benchmark.

## 6 CONCLUSION

In this work, we introduced a framework that integrates discriminative and generative models for time-domain musical source extraction by combining a score-matching diffusion model with a

deterministic model to enhance extraction quality. Our experiments confirmed the hypothesis that incorporating a generative model enhances extraction quality by reconstructing information lost in the mixture due to masking and by reducing source leakage—issues that are common in deterministic approaches.

We overcame the typically slow sampling process of diffusion models by applying Consistency Distillation (CD), which accelerated sampling to speeds comparable to the deterministic model without compromising quality, while also offering significant improvements in exchange for a slight speed trade-off. Notably, this work not only represents the first application of consistency models in the audio waveform domain, showing CD beating diffusion model, but also achieves significant improvements across objective metrics, establishing new state-of-the-art source extraction/separation results on the Slakh2100 dataset.

Our findings highlight the strong potential of mixed deterministic and generative models to advance the field of source extraction and separation, paving the way for future developments in audio processing and related applications. Future work should explore the possibilities of reducing audio segment lengths along with model size of our model, potentially farther decreasing inference times and enabling lightweight, fast, real-time applications for audio source extraction with high-quality results.

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

## A PRELIMINARY

### A.1 SCORE-BASED DIFFUSION MODELS

Score-based diffusion models (Song & Ermon, 2019) are the generative models designed to learn data representations by introducing controlled noise and learning to iteratively remove it from the data. Score-based models are a class of generative models that rely on learning the gradient of the data distribution, known as the score function of the target distribution $p(x)$, namely $\nabla_x \log p(x)$.

Given a data point $x_0$ from the true data distribution $p(x_0)$, the model generates a sequence of noise-corrupted samples $x_t$ by adding Gaussian noise to $x_0$ as $x_t = x_0 + \sigma_t \epsilon$, where $\epsilon \sim \mathcal{N}(0, I)$ represents Gaussian noise, and $\sigma_t$ defines the amount of noise added at time $t$. In score-based diffusion, this process is typically continuous, and with the choice $\sigma_t = t$. Often, as in EDM (Karras

et al., 2022), in training time $\sigma$ can be sampled from a distribution. As $t$ approaches $T$, where $T$ is a sufficiently large value, the process results in isotropic Gaussian noise, with $x_T \sim \mathcal{N}(0, I)$. As described by (Song et al., 2021), the noise removal reverse process in diffusion can be formulated using a probability flow Ordinary Differential Equation (ODE) as follows:

$$\mathrm{d}x_t = \sigma_t \nabla_{x_t} \log p(x_t)\, \mathrm{d}t. \tag{7}$$

By training a neural network to approximate the score function, we can enable a generative process that generates data from noise. For this, the score function is often defined as $\nabla_{x_t} \log p(x_t) = \frac{g_\theta(x_t, \sigma_t) - x_t}{\sigma_t^2}$, where $g_\theta$ is the neural network. The model can be trained with the Denoising Score Matching (DSM) loss, given by:

$$\mathcal{L}(\theta) = \mathbb{E}_{t,x} \left[ \lambda(\sigma_t) \left\| g_\theta(x_t, \sigma_t) - x_0 \right\|_2^2 \right], \tag{8}$$

where $\lambda(\sigma_t)$ is a noise level dependent loss weighting function.

Additionally, for effective training of the network, it is advisable to keep input and output signal magnitudes fixed, for example, to unit variance. To avoid large variations in gradient magnitudes, a common practice is not to represent $g_\theta$ as a neural network directly, but to train a different network, $g'_\theta$, from which $g_\theta$ is derived. EDM describes a preconditioning setup defined as:

$$g_\theta(x_t; \sigma_t) = c_{\mathrm{skip}}(\sigma_t) x_t + c_{\mathrm{out}}(\sigma_t) g'_\theta \big( c_{\mathrm{in}}(\sigma_t) x_t, c_{\mathrm{noise}}(\sigma_t) \big), \tag{9}$$

where $g'_\theta$ is a neural network. The preconditioning parameters regulate the network's skip connection to $x_t$, outputs, inputs, and noise levels respectively as follows:

$$
\begin{aligned}
c_{\mathrm{skip}}(\sigma_t) &= \frac{\sigma_{\mathrm{data}}^2}{\sigma_t^2 + \sigma_{\mathrm{data}}^2}, & c_{\mathrm{out}}(\sigma_t) &= \frac{\sigma_t \cdot \sigma_{\mathrm{data}}}{\sqrt{\sigma_{\mathrm{data}}^2 + \sigma_t^2}}, \\
c_{\mathrm{in}}(\sigma_t) &= \frac{1}{\sqrt{\sigma_t^2 + \sigma_{\mathrm{data}}^2}}, & c_{\mathrm{noise}}(\sigma_t) &= \frac{1}{4} \ln(\sigma_t).
\end{aligned}
\tag{10}
$$

where $\sigma_{\mathrm{data}}$ is the expected standard deviation of the clean data.

In the time of inference, solving an ODE numerically requires approximating the true solution trajectory. In the reverse diffusion process, $\sigma_t$ is a discrete function with finite steps $t \in [1, T]$, such that $\sigma_T = \sigma_{\mathrm{max}}$ and $\sigma_1 = \sigma_{\mathrm{min}}$. (Note: often in the score-based diffusion literature, this parametrization is reversed, with $\sigma_1 = \sigma_{\mathrm{max}}$ and $\sigma_T = \sigma_{\mathrm{min}}$; however, to be consistent with consistency distillation, which also uses this noise scheduler, we adopt the reverse scheduling paradigm.) EDM (Karras et al., 2022) introduced an effective non-linear schedule for time discretization, given by:

$$\sigma_t = \left( \sigma_{\mathrm{min}}^{\frac{1}{\rho}} + \frac{t-1}{T-1} \left( \sigma_{\mathrm{max}}^{\frac{1}{\rho}} - \sigma_{\mathrm{min}}^{\frac{1}{\rho}} \right) \right)^\rho, \tag{11}$$

where $\rho$ controls the curvature or "bend" of the noise schedule.

There are many ODE solvers in the score-based diffusion literature. ODE solvers can be categorized by their order of accuracy, deterministic vs. stochastic nature, etc. For instance, DDIM (Song et al., 2020) and the Euler sampler (Song et al., 2021) correspond to 1st-order deterministic solvers, while EDM (Karras et al., 2022) introduces a deterministic 2nd-order Heun solver. The DPM2 sampler (Lu et al., 2022) uses a 2nd-order deterministic solver, while Euler-Maruyama (Song et al., 2021) represents a first-order stochastic solver.

We denote solvers with $\mathtt{Solver}_k(x_t, \sigma_t, [c]; g_\phi)$, where $c$ is a placeholder for conditioning information, $x_t$ is the starting sample, $g_\phi$ is a neural network trained to approximate the score, $\sigma_t$ is the current noise level, and $k$ is the number of steps the solver performs (solver produces clean sample from pure noise $X_T$ when $k = T$). In the Algorithms 1 and 2 we show examples of deterministic and stochastic solvers used in our experiments.

In Algorithm 1, we present Heun's 2nd order deterministic sampler, which applies a two-step correction mechanism for sampling from a diffusion model. It begins by generating an initial sample

---

**Algorithm 1** Heun's 2$^{nd}$ order deterministic sampler.

---

1: **procedure** HEUN SAMPLER($\text{Solver}_k(x_t, \sigma_t; g_\phi)$)
2:      **sample** $x_T \sim \mathcal{N}\left(\mathbf{0},\ \sigma_{max}^2\ \mathbf{I}\right)$                 $\triangleright$ Generate initial sample at $T$ ($\sigma_T = \sigma_{\max}$)
3:      **for** $t = T, \ldots, 1$ **do**                  $\triangleright$ Solve over $T$ time steps
4:          $d \leftarrow \frac{x_t - g_\phi(x_t, \sigma_t)}{\sigma_t}$                  $\triangleright$ Evaluate derivative at $t$
5:          $x_{t-1} \leftarrow x_t + (\sigma_{t-1} - \sigma_t)d$            $\triangleright$ Euler step from $t$ to $t - 1$
6:          **if** $\sigma_{t-1} \neq 0$ **then**        $\triangleright$ Apply 2$^{nd}$ order correction unless next $\sigma$ is zero
7:             $d' \leftarrow \frac{x_{t-1} - g_\phi(x_{t-1}, \sigma_{t-1})}{\sigma_{t-1}}$          $\triangleright$ Evaluate derivative at $t - 1$
8:             $x_{t-1} \leftarrow x_t + (\sigma_{t-1} - \sigma_t)\frac{1}{2}(d + d')$      $\triangleright$ Explicit trapezoidal rule at $t - 1$
9:          **end if**
10:      **end for**
11:      **return** $x_0$               $\triangleright$ Return noise-free last sample at $t = 1$, gives $x_0$
12: **end procedure**

---

**Algorithm 2** EDM sampler with stochasticity and correction mechanism.

---

1: **procedure** EDM SAMPLER($\text{Solver}_k(x_t, \sigma_t; g_\phi)$)
2:      **sample** $x = x_T \sim \mathcal{N}\left(\mathbf{0},\ \sigma_{max}^2\ \mathbf{I}\right)$      $\triangleright$ Generate initial sample at $T$ ($\sigma_T = \sigma_{\max}$)
3:      $\gamma \leftarrow \min(S_{\text{churn}}/T, \sqrt{2} - 1)$      $\triangleright$ Calculate $\gamma$ value that controls increased noise levels
4:      **for** $t = T, \ldots, 1$ **do**               $\triangleright$ Solve over $T$ time steps
5:          **for** $r = R - 1, \ldots, 0$ **do**          $\triangleright$ Iterate over $R$ correction steps
6:             $\hat{\sigma} \leftarrow \sigma_t \cdot (\gamma + 1)$          $\triangleright$ Calculate temporarily increased noise level $\hat{\sigma}$
7:             **sample** $\epsilon \sim \mathcal{N}(0, I)$             $\triangleright$ Sample noise
8:             $\hat{x} \leftarrow x + \sqrt{\hat{\sigma}^2 - \sigma_t^2}\epsilon$      $\triangleright$ Add noise to move to temporarily increased noise level
9:             $d \leftarrow \frac{\hat{x} - g_\phi(\hat{x}, \hat{\sigma})}{\hat{\sigma}}$           $\triangleright$ Evaluate derivative at $\hat{t}$
10:            $x \leftarrow \hat{x} + (\sigma_{t-1} - \hat{\sigma})d$          $\triangleright$ Euler step from $\hat{t}$ to $t - 1$
11:            **if** $r > 0$ **then**            $\triangleright$ If not last resample step
12:               **sample** $\epsilon \sim \mathcal{N}(0, I)$          $\triangleright$ Sample noise
13:               $x \leftarrow x + \sqrt{\sigma_t^2 - \sigma_{t-1}^2}\epsilon$          $\triangleright$ Renoise
14:            **end if**
15:          **end for**
16:      **end for**
17:      **return** $x$              $\triangleright$ Return noise-free sample after last step $t = 1$
18: **end procedure**

---

at the maximum noise level, $\sigma_{\max}$, and iteratively reduces noise over time using an Euler step followed by a 2nd order correction, based on the trapezoidal rule. This approach improves the accuracy of the generated samples by accounting for the slope at both the current and the next time step, ensuring more precise sampling throughout the process.

Algorithm 2 describes the EDM sampler (also used in MSDM with a little variations). In this sampler, the stochasticity is controlled by the parameter $S_{\text{churn}}$ and it introduces a correction mechanism with $R$ steps for adjusting the trajectories of generated samples. A $\gamma$ value, calculated using $S_{\text{churn}}$, temporarily increases the noise at each step, followed by a resampling procedure that helps correct errors in the diffusion process. The EDM sampler includes $R$ iterations at each time step, resulting in $T \times R$ actual steps in the sampler.

## A.2 CONSISTENCY MODELS

Consistency models (Song et al., 2023) are a novel class of generative models closely related to diffusion models, designed for few-step or even one-step generation. The core idea of consistency models is the concept of self-consistency. Consistency function $g$ directly connects any timestep point of the diffusion trajectory to the trajectory's starting point $g(x_t, t) \to x_\delta$, where $\delta \to 0^+$ represents an infinitesimally small positive value, indicating the very low noise step at the start of the trajectory. This property of the consistency function can also be written as $g(x_t, t) = g(x_{t'}, t'), \forall t, t' \in [\delta, T]$, indicating that any noise level on the given trajectory results in the same output.

The consistency function is typically learned with a deep neural network $g_\omega$ from the data, enforcing the self-consistency property across all timesteps. Consistency models are trained to perform single-step denoising from any time step $t$ to $0$, either through Consistency Distillation (CD) (Song et al., 2023), distilling pretrained teacher diffusion model, or from scratch (Song & Dhariwal, 2024). Similar to inference of score-based models, the time horizon of CD is discretized into $T$ steps and noise levels $\sigma$ are defined following the function from the equation 11 with boundaries $\sigma : [1, T] \rightarrow [\sigma_{\min}, \sigma_{\max}]$. For the consistency function to hold it's *boundary condition*, which is $g_\omega(x_1, \sigma_{\min}) = x_1 = x_0 + \epsilon \sigma_{\min}$ the consistency model $g_\omega$, is parameterized as follows:

$$g_\omega(x_t, \sigma_t) = c_{\text{skip}}(\sigma_t)x_t + c_{\text{out}}(\sigma_t)g'_\omega(x_t, \sigma_t), \tag{12}$$

where $g'_\omega$ is a deep neural network. The preconditioning $c_{\text{skip}}(\sigma_t)$ and $c_{\text{out}}(\sigma_t)$ are differentiable functions, and are usually chosen as:

$$c_{\text{skip}}(\sigma_t) = \frac{\sigma_{\text{data}}^2}{(\sigma_t - \sigma_{\min})^2 + \sigma_{\text{data}}^2}, \quad c_{\text{out}}(t) = \frac{\sigma_{\text{data}}(\sigma_t - \sigma_{\min})}{\sqrt{\sigma_{\text{data}}^2 + t^2}}, \tag{13}$$

which satisfies $c_{\text{skip}}(\sigma_{\min}) = 1$ and $c_{\text{out}}(\sigma_{\min}) = 0$.

In the original CD (Song et al., 2023), the optimization of the consistency model $g_\omega$ is done through minimizing discrepancies between successive estimates of the data at different timesteps. The process begins with a noisy data point $x_t$. The target for the loss is first calculated by taking a single denoising step with an ODE solver, which uses the pretrained score-matching diffusion model $g_\phi$, acting as a teacher, obtaining an estimate of the data at the $t-1$ step, $\hat{x}_{t-1}^\phi = \texttt{Solver}_1(x_t, \sigma_t; g_\phi)$. Then, the stop-gradient EMA updated copy of the student model performs a step from $t-1$ to $0$. The estimate for the loss is calculated by the student model $g_\omega$, which directly jumps from $t$ to $0$. The loss is then defined as follows:

$$\mathcal{L}_{\mathcal{CD}}(\omega) = \mathbb{E}_{x,t}\Big[d\big(\underbrace{g_{\texttt{sg}(\omega)}(\hat{x}_{t-1}^\phi, \sigma_{t-1})}_{target}, \underbrace{g_\omega(x_t, \sigma_t)}_{estimate}\big)\Big], \tag{14}$$

where $\texttt{sg}(\theta)$ denotes the stop-gradient running EMA updated $g_\omega$ during optimization, and $d(\cdot, \cdot)$ is any function used to measure the distance between the model's predictions.

The Consistency Trajectory Model (CTM) (Kim et al., 2024) presented a unified approach that integrates score-based and consistency distillation models by allowing both infinitesimally small and long-step jumps along the Probability Flow ODE trajectory. CTM trains a neural network $g_\omega(x_t, \sigma_t, \sigma_{t'})$ to predict the solution of the ODE from an initial time $t$ to a final time $t'$, with $t' < t$, enabling any-step-to-any-step predictions. The preconditioning for the network is as follows:

$$g_\omega(x_t, \sigma_t, \sigma_{t'}) = \frac{\sigma_{t'}}{\sigma_t}x_t + \left(1 - \frac{\sigma_{t'}}{\sigma_t}\right)g'_\omega(x_t, \sigma_t, \sigma_{t'}), \tag{15}$$

where $g'_\omega$ is a neural network, and thus $g_\omega$ automatically satisfies the initial boundary condition. Using same discrete noise scheduling, we have $g_\omega(x_1, \sigma_{\min}, \sigma_{\min}) = x_1 = x_0 + \epsilon \sigma_{\min}$.

For the distillation loss in CTM, the teacher model first takes steps from $t$ to $u$, where the noise level $u$ is randomly chosen between the start and end as $u \in [t', t)$. Thus, we obtain an estimate $\hat{x}_u^\phi = \texttt{Solver}_{t-u}(x_t, \sigma_t; g_\phi)$. Then, the stop-gradient student model proceeds and jumps from $u$ to $t'$, followed by a step to $0$ to calculate the target. For the estimate, the student model directly transitions from $t$ to $t'$, and similarly proceeds to $0$ using the stop-gradient model. The optimization of the consistency model $g_\omega$ is done through minimizing the loss, which is calculated as follows:

$$\mathcal{L}_{\text{CTM}}(\omega) = \mathbb{E}_{t,t',u}\Big[d\big(\underbrace{g_{\texttt{sg}(\omega)}(g_{\texttt{sg}(\omega)}(\hat{x}_u^\phi, \sigma_u, \sigma_{t'}), \sigma_{t'}, 0)}_{target}, \underbrace{g_{\texttt{sg}(\omega)}(g_\omega(x_t, \sigma_t, \sigma_{t'}), \sigma_{t'}, 0)}_{estimate}\big)\Big]. \tag{16}$$

Additionally, in the CTM framework, two auxiliary losses are used. First, the model is trained to approximate infinitesimally small neural jumps when $t' \rightarrow t$, using the DSM loss:

$$\mathcal{L}_{\text{DSM}}(\omega) = \mathbb{E}_t \left[ \|x_0 - g_\omega(x_t, \sigma_t, \sigma_t)\|_2^2 \right]. \tag{17}$$

The second auxiliary loss is introduced by incorporating an additional GAN component into the system, introducing the GAN loss, and making the final loss a weighted sum of these terms:

$$\mathcal{L}(\omega, \eta) = \mathcal{L}_{\text{CTM}}(\omega) + \lambda_{\text{DSM}} \mathcal{L}_{\text{DSM}}(\omega) + \lambda_{\text{GAN}} \mathcal{L}_{\text{GAN}}(\omega, \eta), \tag{18}$$

where $\lambda_{\text{DSM}}$ and $\lambda_{\text{GAN}}$ are adaptive weights to stabilize training by balancing the gradient scale of each term.

## B    ADDITIONAL EXPERIMENTAL DETAILS

### B.1    DETERMINISTIC MODEL

For our deterministic Model, we adopted U-Net architecture consisting of 1D CNNs with self-attention and enhanced convolutional blocks. The U-Net, originally introduced for biomedical imaging (Ronneberger et al., 2015), consists of convolutional layers in both the encoder and decoder, connected by skip connections.

We adopted the hyperparameter setting used by MSDM (Mariani et al., 2024), with the modification of using a single channel instead of four and using U-Net without a diffusion part. The model implementation is based on the publicly available repository `audio-diffusion-pytorch/v0.0.432`[1]. Our U-Net encoder consists of six layers, each containing two convolutional ResNet blocks. The first three layers do not incorporate attention, while the last three include multi-head attention with 8 heads and 128 attention features. The downsampling factor is 4 in the first three layers and 2 in the last three layers. The number of channels in the encoder is [256, 512, 1024, 1024, 1024, 1024]. The bottleneck includes a ResNet block, followed by a self-attention mechanism, and another ResNet block, all maintaining 1024 channels. The decoder mirrors the encoder, following a symmetric structure in reverse.

### B.2    DIFFUSION MODEL

Our diffusion model uses the same architecture and set of hyperparameters for the backbone U-Net as in the deterministic Model. Adopting preconditioning from the EDM framework, the standard deviation of the data was set to $\sigma_{\text{data}} = 0.2$, and during training, $\sigma$ was sampled from a log-normal distribution with $\ln(\sigma) \sim \mathcal{N}(P_{\text{mean}}, P_{\text{std}}^2)$, where $P_{\text{mean}} = -3$ and $P_{\text{std}} = 1$. The loss weighting was performed using $\lambda(\sigma_t) = \frac{1}{c_{\text{out}}(\sigma_t)^2}$, where $c_{\text{out}}(\sigma_t)$ is taken from equation 10.

For inference, we found the EDM sampler, given in Algorithm 2, to be most effective for our diffusion model. In our experiments, we set the minimum noise parameter to $\sigma_{\text{min}} = 0.0001$ and found that $\rho = 9$ and $S_{\text{churn}} = 20.0$ were optimal. The best results for our diffusion model were achieved with $T = 5$ sampling steps, $R = 2$ correction steps, and $\sigma_{\text{max}} = 0.01$. Additionally, the parameter $\sigma_{\text{max}}$ was varied between 0.001 and 80, and the impact of these values can be observed in Figure 2 in the results section.

### B.3    CONSISTENCY DISTILLATION

The architecture and hyperparameter settings for the consistency model exactly follow those of our diffusion model, with differences in the training procedure settings. The $\sigma$-scheduling for the CD loss fully adhered to the discrete form in equation 11. We set $\sigma_{\text{min}} = 0.0001$, $\sigma_{\text{max}} = 10.0$ and $\rho = 9.0$ for the CD training. The $\sigma$-schedule for the DSM loss, on the other hand, followed a half-lognormal distribution with $P_{\text{mean}} = -3$ and $P_{\text{std}} = 1$, where half of the batch in each iteration was sampled from this lognormal distribution, and the other half was sampled by uniformly selecting $t$ and calculating $\sigma_t$ from equation 11. For distilling the learning trajectories, a Heun's 2nd-order deterministic ODE solver, shown in Algorithm 1, was used with the teacher model. The number of

---

[1]`https://github.com/archinetai/audio-diffusion-pytorch/tree/v0.0.43`

ODE steps was uniformly sampled from $h \sim U[1, 17]$. We experimented with weight scheduling for both the $\mathcal{L}_{CD}$ and $\mathcal{L}_{DSM}$ losses. We found that using uniform weighting for $\mathcal{L}_{CD}$ and applying $\lambda(\sigma_t) = \frac{1}{c_{out}(\sigma_t)^2}$ (as in the diffusion model from equation 10) for $\mathcal{L}_{DSM}$ provided the best training stability and performance. Additionally, we tested adaptive balance weighting between these losses, as defined in equation 6, and found that setting $\lambda_{DSM} = 1$ was optimal for our CD training.

For inference, we employed the *onestep* and *multistep* algorithms, as described in Song et al. (2023). However, as mentioned earlier, instead of initializing the diffusion process with pure noise $x_T \sim \mathcal{N}(0, \sigma_{max}^2\mathbf{I})$, we start with a sum of noise and the output $\hat{x}_s^{det}$ of the deterministic model as: $x_T = \hat{x}_s^{det} + \sigma_{max} \cdot \epsilon$, where $\epsilon \sim \mathcal{N}(0, \mathbf{I})$. As evident from our results, small values of $\sigma_{max} \in [0.001, 0.5]$ resulted in the best performance during the inference process.

Table 3: **Comparison of results of out model with baseline models on MUSDB18 test set (Rafii et al., 2017).** We report the SI-SDR$_I$ values in dB (higher is better).

| Model | Trained on | Bass | Drums | Other | Vocals | All |
|---|---|---|---|---|---|---|
| Demucs v2 (Mariani et al., 2024) | MUSDB | 13.28 | 11.53 | 8.59 | 16.80 | 12.55 |
| MSDM (Mariani et al., 2024) | MUSDB | 4.87 | 3.28 | 1.97 | 6.83 | 4.24 |
| Demucs (as reported in (Manilow et al., 2022)) | MUSDB | 6.80 | 5.25 | -1.30 | 3.90 | 3.66 |
| Demucs + Gibbs (as reported in (Manilow et al., 2022)) | MUSDB | 9.00 | 7.15 | 1.00 | 11.30 | 7.61 |
| MSDM (Mariani et al., 2024) | Slakh | -0.83 | -0.94 | - | - | -0.88 |
| Deterministic Model | Slakh | 3.55 | 5.19 | - | - | 4.37 |
| Diffusion | Slakh | 3.50 | 5.15 | - | - | 4.32 |
| CD 1 step | Slakh | 3.54 | 5.14 | - | - | 4.34 |
| CD 2 step | Slakh | 3.83 | 5.04 | - | - | 4.43 |
| CD 4 step | Slakh | 3.70 | 4.90 | - | - | 4.30 |
| Deterministic Model | MUSDB | 9.93 | 8.19 | 4.59 | 12.32 | 8.75 |

## C RESULTS ON MUSDB DATASET

Reported in the Table 3, we compared the performance of our model with MSDM, Demucs v2, and Demucs + Gibbs on the MUSDB18 (Rafii et al., 2017) test set, reporting SI-SDR$_I$ values (dB). First, we evaluated our model trained on Slakh2100 without any training or fine-tuning on MUSDB18, reporting results only for bass and drums, as our model is not trained on the other and vocals categories in this setting. We found that our model, when evaluated on the MUSDB18 test set, outperformed MSDM trained on Slakh2100 and even MSDM trained on MUSDB18 (as reported in their paper). However, we observed that the diffusion and CD extensions of our model did not show any additional improvement over the deterministic model on the MUSDB dataset.

Next, we trained our deterministic model on the MUSDB18 dataset and found that it outperformed both Demucs and Demucs + Gibbs as reported in Manilow et al. (2022). However, our deterministic model did not outperform Demucs v2, as reported in the MSDM paper. Additionally, we did not observe any improvement from the diffusion and CD methods over the deterministic model when further trained on the MUSDB dataset. The results were similar to those of the deterministic model, and thus we do not report them here.

While our model demonstrated reasonable performance on the MUSDB18 dataset, the limited size of MUSDB18 (10 hours) compared to the significantly larger Slakh2100 dataset (145 hours) likely hindered the full potential of our diffusion-based approach. We believe the smaller dataset size was insufficient to fully leverage the capabilities of our diffusion based method and plan to investigate possibilities for adapting our approach to better accommodate smaller datasets in future work.

