# OpenReview forum: "Improving Source Extraction with Diffusion and Consistency Models"
_ICLR.cc/2025/Conference — Submitted to ICLR 2025_

### Official Review · Reviewer_3PvU · 2024-11-03

**Soundness:** 1
**Presentation:** 2
**Contribution:** 2
**Rating:** 3
**Confidence:** 4

**Summary:**

This paper presents a way to use score-based diffusion models to improve music source separation. By adding a score-based model the performance improves, and consistency distillation improves the model's inference speed and sometimes further improves the performance.

**Strengths:**

1. This paper demonstrates a new training strategy for source separation models of high SDR and efficiency, which might be beneficial to the community.
2. The improvement of SDR acquired by consistency distillation is pretty interesting. It can open up possibilities for future research.

**Weaknesses:**

Experiments: The experiments are very weak. Slahk is only a synthetic dataset created by MIDI synthesizers, which generalizes poorly to real-world recordings (as shown in table 3), so a state-of-the-art SDR on Slahk is less meaningful.

The author says that the diffusion model's performance on real-world recordings (MusDB18) is not improved compared to the deterministic model because of the dataset size, and did not report the results in table 3. This raises concerns about the author's conclusion. It would be more appropriate to report whatever the authors got when training the diffusion model and the consistency model using MusDB18. Also, there are many ways to mitigate the dataset size issue by (1) data augmentation like pitch shifting etc [1], (2) combining with other datasets like MoisesDB, and/or (3) using transfer learning.

If the author's claim only holds on a synthetic dataset, it would be highly questionable.

Contributions: It seems that the paper only applies an existing methodology to a specific task. The contribution might be questionable. The idea of using the diffusion model in source separation is not novel either, as cited by the authors.

[1] Défossez, A. (2021). Hybrid spectrogram and waveform source separation. arXiv preprint arXiv:2111.03600.

**Questions:**

1. The improvement acquired by distillation is questionable. In section 4.2, the model trained by CD uses different hyperparameters (i.e., learning rate) compared to the diffusion model. Could the improvement be caused by learning rate annealing?

2. In 3.2, how is $\bar{x}^\textrm{det}_s$ calculated? It seems in figure 1 that it is from the layers of the frozen u-net. But specifically, which layers? (Also typo in figure 1: diffusion extension?)

3. The title uses the word "source extraction" which seems a little bit strange. Even if each model only extracts one stem at a time, it is usually described as "source separation."

---

> ### Author Response · Authors · 2024-11-21
>
> ## Response to Strengths
>
> We thank the reviewer for their positive feedback and for recognizing the contributions of our work. However, we wouldn’t necessarily describe our method as a “training strategy” for source separation. Instead, we see our work as more conceptual, bridging discriminative and generative methodologies in source separation.
> We appreciate the acknowledgment of the use of consistency distillation (CD). We share the view that this approach opens up exciting possibilities for future research, particularly in accelerating diffusion-based models, and we hope that our work inspires further exploration and innovation in this area.
>
> ## Response to Weaknesses
>
> ### Experiments and Dataset Generalization
> We acknowledge the reviewer's concerns about the use of the Slakh2100 dataset, a synthetic dataset created from MIDI synthesizers. While we agree that Slakh2100 may not generalize as well to real-world recordings, we chose this dataset because it is widely used in the community, allows for direct comparisons with existing baselines, and is large enough to support the data-hungry nature of diffusion models. We believe that demonstrating state-of-the-art performance on Slakh2100 is meaningful within the context of these controlled conditions. As we already stated in the answers to the other reviewers, our goal was not to build the best in the world source separation model but to test the application of generative diffusion models for potential improvement over deterministic methods, with consistency distillation (CD) employed to accelerate performance.
> Regarding Table 3, we clarified in the appendix that the diffusion and consistency models trained on Slakh2100 did not perform well on MUSDB18 because they were not trained on this dataset. The deterministic model trained on MUSDB18 was reported in Table 3 and outperformed several baselines. We agree that reporting the results for the diffusion model trained directly on MUSDB18 would strengthen the analysis, and we will include these results in a future revision. We will add the number in the following revision.
>
> The reviewer suggested several ways to mitigate the dataset size issue, including data augmentation, combining datasets, and transfer learning. These are excellent suggestions, and we plan to explore these techniques in future work. However, our primary goal in this study was to demonstrate the conceptual feasibility of integrating deterministic and generative methods for source separation, rather than addressing all challenges related to dataset size or domain generalization. Again we were not trying to build the best source separation model!
>
> ### Contribution and Novelty
> We understand the reviewer's concern about the contribution of the paper. While the idea of applying diffusion models to source separation is not novel, as noted in the paper, the main contribution lies in combining deterministic and generative models within a unified framework and demonstrating that this approach can improve performance. Furthermore, the use of consistency distillation (CD) in this context is novel, as it significantly accelerates the inference process of diffusion models while maintaining or improving separation quality.
>
> We agree diffusion models are not novelty in source separation but diffusion used for the refinement of separation of deterministic models are indeed novel.

---

> ### Author Response · Authors · 2024-11-21
>
> ## Response to Questions
>
> ### Improvement Acquired by Distillation
> We understand the concern regarding the improvement acquired by consistency distillation (CD). While it is true that the CD model uses a different learning rate compared to the diffusion model, we do not believe the improvement is solely due to learning rate annealing. The primary benefit of CD comes from distillation of the diffusion model, and the distillation procedure is fundamentally different from the main training procedure of diffusion model. The distillation process, where the model learns to jump from high-noise levels to clean samples, cannot be considered a continuation of the denoising training of the diffusion model. Therefore, discussing annealing in this context is not relevant.
>
> In CD, student models are typically initialized with the weights of the original diffusion model, and using a low learning rate is standard practice. Please refer to the CD [1] and CTM [2] papers for further details.
>
> [1] Yang Song, Prafulla Dhariwal, Mark Chen, and Ilya Sutskever. Consistency models. In International
> Conference on Machine Learning, ICML 2023, 23-29 July 2023, Honolulu, Hawaii, USA, volume
> 202 of Proceedings of Machine Learning Research, pp. 32211–32252. PMLR, 2023
>
> [2] Dongjun Kim, Chieh-Hsin Lai, Wei-Hsiang Liao, Naoki Murata, Yuhta Takida, Toshimitsu Uesaka, Yutong He, Yuki Mitsufuji, and Stefano Ermon. Consistency trajectory models: Learning probability flow ODE trajectory of diffusion. In The Twelfth International Conference on Learning
> Representations, ICLR 2024, Vienna, Austria, May 7-11, 2024, 2024.
>
> ### Calculation of $ \bar{x}^{sdet} $
> Thank you for raising this question. In Figure 1, $ \bar{x}^{sdet} $ is calculated as an intermediate output from the frozen U-Net layers of the deterministic model. Specifically, it is obtained from the output of every layer in both the encoder and decoder, and these outputs are used as conditioning for the corresponding layers in the diffusion U-Net. This conditioning is implemented by simply adding the outputs of the deterministic U-Net layers to the outputs of the diffusion U-Net layers at the same level.
>
> We will clarify this in Section 3.2 and update the figure caption to reflect this explanation accurately. Additionally, we will correct the typo in Figure 1 ("diffusion extantion" → "diffusion extension").
>
> ### Use of "Source Extraction" in the Title
> Thank you for the question regarding the use of "source extraction" in the title. While "source separation" is the standard term in the field, we chose "source extraction" to emphasize the one-at-a-time extraction process used in our method. Our model performs source extraction rather than source separation, as the task of source separation is only complete when all four sources are extracted. We provided a specific explanation of this distinction in the introduction section (lines 023-026).
>
> We believe that "source extraction" is also a well-established term in the literature, and its usage here accurately reflects the methodology employed in our work.

---

> ### Comment · Reviewer_3PvU · 2024-11-23
>
> Thanks for the clarification. The author's rebuttal helps clarify some concerns, but it does not help resolve the main issue.
>
> The author says that "our goal was not ... but to test the application of generative diffusion models for potential improvement over deterministic methods," while claiming "the diffusion model did not show an improvement over the deterministic model on MUSDB." It is possible that MusDB is a small-scale dataset so the diffusion model cannot be trained well, but this fact still undermines the paper's conclusion. This also shows a potential drawback that the proposed model is more data hungry.
>
> I currently could not find enough evidence to support the acceptance of the paper, and I would like to refer to the revised version if possible.

---

### Official Review · Reviewer_2aY5 · 2024-11-03

**Soundness:** 3
**Presentation:** 4
**Contribution:** 2
**Rating:** 5
**Confidence:** 4

**Summary:**

The paper proposes a novel method of source extraction i.e. the problem of isolating or extracting a specific sound source from a mixture of audio signals. The authors propose a 2-stage pipeline consisting of a discriminative model and a generative model. The discriminative model makes a preliminary estimation of the extracted source x_s given the mixture signal x and the instrument label s. Then, x_s is used as a conditioning input to the diffusion-based generative model to produce a refined version of the extracted source for the instrument s. As the next step, the authors employ knowledge transfer to train a consistency model and speed up the sampling process. The method is tested on data from Slakh2100 and MUSDB18 datasets demonstrating clear advantages over several other models including MSDM and ISDM.

**Strengths:**

- The paper is well written and easy to follow.
- The problem addressed by the method is challenging and important for many applications in music information retrieval.
- The method shows significant improvement over baselines on Slakh2100.
- Authors successfully train a consistency model and considerably speed up their pipeline.

**Weaknesses:**

- Unfortunattely, the full pipeline has demonstrated no improvement on non-MIDI data from MUSDB18 compared to the discriminative part and baseline Demucs v2. The experiment doesn’t provide full understanding of performance of the method on non-MIDI data;
- Source separation problem is treated as a sequence of source extraction problems for each instrument in the mix making either memory or runtime grow linearly with the number of instruments;
- Role of the diffusion-based and/or consistency model used in the method is to refine the outputs of the discriminative model. Both models have U-Net architecture with no essential design changes. Generally, the method doesn’t introduce original design choices.

**Questions:**

Comparison with Demucs provided in Table 1 doesn’t seem fair as Demucs was trained on much less data. Moreover, Demucs was trained on non-MIDI data and tested on MIDI data while the proposed model was trained and tested on Slakh2100. Did you consider retraining Demucs on Slakh2100 for better comparison?

---

> ### Author Response · Authors · 2024-11-21
>
> ## Response to Strengths
>
> We thank the reviewer for their positive feedback and for recognizing the strengths of our work. We are pleased that the paper is considered well-written and easy to follow, as clarity was a key priority for us when presenting this challenging and important problem in music information retrieval. We appreciate the acknowledgment of the significant improvements our method demonstrates over baselines on the Slakh2100 dataset. Additionally, we are glad that our successful implementation of consistency distillation and its contribution to significantly speeding up the pipeline have been noted. This validation reinforces the relevance and impact of our approach.
>
>
> ## Response to Weaknesses
>
> ### Performance on Non-MIDI Data
> As noted in our response to another reviewer, Table 3 (lines 830-833) presents the performance of the diffusion and CD extensions trained on Slakh2100 and evaluated on MUSDB. Unsurprisingly, these models did not perform well on MUSDB since they were not trained on it, which explains their inability to outperform the baselines. However, the row below (line 834) shows the results for our deterministic model trained directly on MUSDB. This model surpasses several baselines, including Demucs (line 827), Demucs+Gibbs (line 828), and MSDM trained on MUSDB (line 826). The only baseline it does not outperform is Demucs2 (as reported in the MSDM paper), which is not the Demucs1 used as a baseline in our other experiments. Demucs2 is not strictly a deterministic method and leverages additional techniques to boost performance.
>
> The main claim we make in the appendix is not that the deterministic model failed to outperform the baselines, but rather that the diffusion model did not show an improvement over the deterministic model on MUSDB as it did on Slakh2100. We did not present results for the diffusion model trained on MUSDB, which proved to be a mistake (as we realize now) and may have caused confusion for the readers. We acknowledge that this distinction was not communicated clearly, and we will revise the manuscript to include more detailed explanations and provide the diffusion model's results for added clarity.
>
> ### Memory and Runtime Scaling with the Number of Instruments
> We appreciate and understand the linear scaling of memory and runtime with the number of instruments. However, we don’t see this as a weakness of the system. This is simply a domain and approach choice we decided to pursue. We see source extraction as a more applicable application than source separation because, theoretically, it is easier to ask the model to provide an extracted source. Often, performing source separation is impractical because the number and types of sources might not be known. Extraction additionally enables a modular design and simplifies handling specific sources that are needed. Therefore, performing source separation or source extraction one by one is just a different approach to a similar family of problems, and we don’t see how this can be considered a weakness of the work.
>
> As for memory and runtime, we believe Table 2 clearly shows that even though our method requires making 4 extractions sequentially, it is still competitive with baseline methods and, in most cases, even outperforms them. Thus, we respectfully disagree with the reviewer on this point.
>
> ### Lack of Original Design Choices
> We understand the reviewer's concern regarding the lack of essential design changes in the diffusion and discriminative models, which both use U-Net architectures. However, we must stress that using the same U-Net architecture was a deliberate design choice, as it allows for dimensional compatibility for layer-by-layer conditioning from the deterministic model to the diffusion model. Additionally, this choice provides a neutral baseline condition for testing our main hypothesis: that deterministic and generative models can complement each other to refine audio separation quality.
>
> Our focus in this work was not on introducing novel architectures but on demonstrating how generative diffusion models can refine outputs from deterministic models and how consistency distillation can accelerate this process. Introducing new design choices would have made the main point of our study obsolete and shifted the focus elsewhere. We see this deterministic+generative cascaded approach itself as a design choice, which our experiments have proven successful.
>
> We may consider exploring different model design choices in future studies to further improve audio quality and speed.

---

> > ### Comment · Reviewer_2aY5 · 2024-11-26
> >
> > Thank you for your response. Unfortunattely, I'm still not convinced that the contribution of this paper is strong enough. The experiments are not satisfying as noted by all reviewers and the usage of diffusion-based models for post-processing and enhancement is not novel.

---

> ### Author Response · Authors · 2024-11-21
>
> ## Response to Questions and Suggestions
>
> ### Demucs metrics
> We thank the reviewer for raising this concern. However, we would like to clarify that the comparison with Demucs in Table 1 is indeed fair. The numbers for Demucs were taken directly from the MSDM paper [1], which reported results from Demucs+gibbs paper [2] that explicitly stated they had trained Demucs (plian Demucs, version 1, no tricks used) on Slakh2100. Therefore, the Demucs results used in our comparison represent a model trained and tested on Slakh2100, aligning with the setup of our proposed method.
>
> Retraining Demucs ourselves was not necessary since the reported numbers were already available under identical conditions. We made sure to adopt exactly the same procedure for evaluation as our baselines to make comparison fair and direct. We explain this over the lines 285-289 . This ensures that the comparison is valid and adheres to the same dataset and evaluation procedures.
>
> [1] Mariani, Giorgio, et al. "Multi-source diffusion models for simultaneous music generation and separation." in Proc. ICLR (2024).
>
> [2] E. Manilow, C. Hawthorne, C. -Z. A. Huang, B. Pardo and J. Engel, "Improving Source Separation by Explicitly Modeling Dependencies between Sources," ICASSP 2022 - 2022 IEEE International Conference on Acoustics, Speech and Signal Processing (ICASSP), Singapore, Singapore, 2022, pp. 291-295,

---

> ### Author Response · Authors · 2024-11-26
>
> Thank you for your comment. Regarding your statement that "the usage of diffusion-based models for post-processing and enhancement is not novel," we would greatly appreciate it if you could provide references to support this claim. As the authors of this paper, we are convinced that this is the first instance of using diffusion models for for post-processing and enhancement purpose, at least within this domain. If there are indeed prior works, we would be grateful to include them in the revised version of our paper to ensure a comprehensive related works section.

---

### Official Review · Reviewer_Tp1m · 2024-11-04

**Soundness:** 2
**Presentation:** 3
**Contribution:** 2
**Rating:** 3
**Confidence:** 4

**Summary:**

This paper introduces a method for musical source separation that integrates a score-matching diffusion model with consistency distillation, addressing the typically slow iterative sampling of diffusion-based methods. The authors first train a deterministic model for source separation and then enhance its output by training a diffusion model on top of it. Evaluated on the Slakh2100 dataset, the proposed approach demonstrates improved performance in instrument separation compared to baseline methods.

**Strengths:**

The integration of score-matching diffusion with consistency distillation for source separation presents an interesting approach, particularly with conditioning on the ground truth source stems during training and on $\hat{x}_s^\text{det}$ during inference, which aligns well with the demands of source separation. This method demonstrates objective improvements over baseline systems, specifically achieving higher performance on the Si-SDR metric when evaluated on the Slakh2100 dataset.

**Weaknesses:**

The experimental setup in this paper is limited, as the authors primarily rely on the Slakh2100 dataset, which consists of MIDI-synthesized data, raising concerns about the generalizability of the results to real-world music. Additionally, the evaluation focuses solely on the Si-SDR metric, which, while relevant, does not fully capture separation “quality”. Other metrics, such as SDR, SIR, and SAR, could offer a more comprehensive analysis of the model’s performance. Although the authors briefly report results on the MUSDB dataset in the appendix, they only provide results of the Deterministic Model, noting that the diffusion and CD extensions showed no improvement, which they attribute to dataset size limitations. However, many source separation models successfully benchmark on MUSDB, challenging this explanation. Incorporating additional datasets, such as MoisesDB [1] and MUSIC [2], would have allowed for a larger and more diverse dataset, strengthening the method’s robustness and improving its generalizability claims.

If the authors aim to position their method as state-of-the-art (SoTA) in source separation, they should have tested it more comprehensively across varied datasets and metrics to establish robustness and generalizability. Alternatively, if the focus is to demonstrate the effectiveness of CD, the authors should have either extended their analysis to other data distributions or thoroughly examined why CD works on specific datasets and not others, rather than attributing its limitations solely to dataset size. Unfortunately, the authors have not achieved either, and reliance on a single dataset and limited metrics does not adequately demonstrate the method’s potential, especially for a top-tier conference like ICLR.

[1] Pereira, Igor, et al. "Moisesdb: A dataset for source separation beyond 4-stems." *in Proc. ISMIR* (2023).

[2] Zhao, Hang, et al. "The sound of pixels." *Proceedings of the European conference on computer vision (ECCV)*. 2018.

**Questions:**

- Why was a 1D architecture preferred over 2D representations (e.g., complex spectrograms or mel spectrograms with vocoder models)? Are there any findings that suggest better performance with the 1D time-domain approach?
- dataset - What does the percentile of instrumental “presence” mean?
- Baseline systems
    - Were the baseline systems trained from scratch using the same dataset setup as the proposed approach?
    - Table 1: the baseline results are identical to those in [3] (Table 3). Did the authors ensure that the evaluation conditions are consistent?
    - Table 2: the inference time and parameter count for Demucs differ from [3] (Table 4). Could the authors clarify these discrepancies?
- line 307 - can we say CD improves both performance and quality? which metric was used to assess the quality?
- The reviewer didn’t fully get the intention of Figure 2, as it presents results that were observed from previous consistency distillation papers. Why compare CD and Diff according to each denoising steps? Why did the authors use denoising steps of Diff to 5 where it could be denoised with larger number of steps?

[3] Mariani, Giorgio, et al. "Multi-source diffusion models for simultaneous music generation and separation." *in Proc. ICLR* (2024).

-----
- typo @Figure 1 (b) caption: “extantion”
- typo @line 33: “…, as illustrated by the “cocktail party effect” …” → quotation mark
- typo @line 171: “This process continuous until …” → continues
- typo @line 179: “…, CD is a designed as discrete process …”
- Figure 2: better to change the legend “U-Net” to “Deterministic”. use different line style for CD and Diff

---

> ### Author Response · Authors · 2024-11-21
>
> ## Response to Strengths
>
> We thank the reviewer for recognizing the strengths of our method, particularly the integration of score-matching diffusion with consistency distillation for source separation. We indeed consider this integration to be our main contribution, especially the refinement of the output of the deterministic model, \( x^{sdet} \) by the diffusion model, as the reviewer correctly noted. The objective improvements our method demonstrates over baseline systems, particularly the higher performance on the Si-SDR metric when evaluated on the Slakh2100 dataset, validate that our method is effective and that our contributions are both impactful and relevant to the field.
>
> ## Response to Weaknesses
>
> ### Experimental Setup, Dataset Generalizability, and Metrics
> We understand the reviewer’s concern regarding the use of the Slakh2100 dataset and its implications for the generalizability of our results to real-world music. We acknowledge that Slakh2100 consists of MIDI-synthesized data. However, as diffusion-based generative models are famously data-hungry, no other source separation dataset, apart from Slakh2100, is large enough to work well with diffusion models. Additionally, we prioritized comparability with baselines, which also rely on Slakh2100 and MUSDB datasets, making these two datasets the most practical choices for our study.
>
> Regarding the reliance on the Si-SDR metric, this decision was again motivated by the need for direct and fair comparisons with baselines, which also use Si-SDR as their primary evaluation metric. We followed the exact procedure from our baselines, described in lines 285-289 of the paper, to ensure comparability.
>
> Concerning the MUSDB results in the appendix, we believe there may have been some misunderstanding. In Table 3 (lines 830-833), we show the performance of the diffusion and CD extensions trained on Slakh2100 and tested on MUSDB. As expected, these models did not perform well on MUSDB because they were not trained on it, which explains why they did not outperform the baselines. However, in the row below (line 834), we show results for our deterministic model trained directly on MUSDB. This model outperforms several baselines, including Demucs (line 827), Demucs+Gibbs (line 828), and MSDM trained on MUSDB (line 826). The only baseline it does not outperform is Demucs2 (as reported in the MSDM paper), which is not purely a deterministic method and includes additional enhancements for improved performance.
>
> The claim we make in the appendix is not that our deterministic model failed to outperform baselines, but rather that our diffusion model did not show the same level of improvement over the deterministic model on MUSDB as it did on Slakh2100. We acknowledge that this might not have been sufficiently clear, and we will revise the manuscript to include more explicit explanations and provide the diffusion model's numbers for added clarity.
>
> ### Positioning the Method and Focus on CD
> We believe there may have been some misunderstanding regarding the aims of our paper. We do not claim to have developed a state-of-the-art (SoTA) method for source separation. Instead, our focus is on addressing a narrower problem: demonstrating that deterministic and generative methods can work together to achieve improved performance. Our main contribution is showing that combining deterministic and generative approaches yields better results compared to using deterministic methods alone.
>
> Regarding CD, our goal was primarily to demonstrate its potential in the context of source separation as a method to accelerate performance. We see our work as a proof of concept, showing the feasibility of integrating deterministic and generative approaches, and see CD as a method to accelerate diffusion process.
> We will revise the manuscript to clarify the scope and goals of our study and ensure they are better aligned with the reviewer's/reader’s expectations.

---

> > ### Author Response · Authors · 2024-11-21
> >
> > ## Response to Questions and Suggestions
> >
> > ### 1D Architecture vs. 2D Representations
> > We chose a 1D time-domain architecture over 2D (spectrogram-based) methods primarily as a domain-specific choice rather than as an optimization for the best-performing model. As stated earlier, our goal was not to build the best in the world source separation model (in 1D or 2D) but to test the application of generative diffusion models for potential improvement over deterministic methods, with consistency distillation (CD) employed to accelerate performance.
> >
> > The decision to operate in the waveform domain was somewhat arbitrary, motivated by its ability to directly model temporal dynamics without intermediate transformations like spectrograms, which can introduce artifacts during transformation and inversion. Additionally, we were curious to explore the application of CD outside of image data, directly in the waveform domain, which served as a secondary motivation for this choice.
> >
> > ### Dataset - Instrumental “Presence” Percentile
> > The "percentile of instrumental presence" refers to the percentage of mixtures that include each instrument relative to the total number of mixtures. For example, a 94% drum presence means that 94% of the 12-second mixture segments contain a non-silent drum track. This metric provides a quantitative measure of an instrument’s prominence within the dataset.
> >
> > ### Baseline Systems
> > - **Training from Scratch:** All baseline systems were trained or evaluated using the same dataset and setup as our proposed approach. We ensured identical datasets, metrics, and evaluation procedures for fair comparison. Our codebase was built on top of MSDM [1], and we ensured that the dataset and evaluation pipelines matched theirs exactly. That said, we did not retrain all baseline models ourselves (as not all of them are open-source) and instead relied on the numbers reported in their respective papers.
> > - **Consistency of Evaluation Conditions (Table 1):** We confirm that the baseline results in Table 1 are consistent with those reported in the MSDM [1] paper. We adopted the same evaluation conditions, datasets, and metrics to maintain comparability.
> > - **Discrepancies in Table 2:** We appreciate the reviewer pointing out discrepancies in inference time and parameter count for Demucs. We anticipated this question.
> >
> > During our research, we noted discrepancies between the MSDM paper [1] and the original Demucs paper [2] (Table 5), where model sizes for Demucs are provided. Because of these differences, we verified the model sizes and generation speeds ourselves. We downloaded the pre-trained Demucs model and ran experiments, observing that the reported model size differed from MSDM’s claims, though the generation time was fairly close. These discrepancies are reflected when comparing Table 2 from our paper with Table 4 from MSDM.
> >
> > We suspect the MSDM paper may have mistakenly reported the size of a of Demucs.
> >
> > ### CD and Quality Assessment (Line 307)
> > CD improves quality as quantified by Si-SDR. In our paper, "quality" was assessed using Si-SDR, which indirectly reflects audio quality through signal fidelity. The performance improvement is inherent to CD’s design, as it generates audio of equal or better quality in 1–4 steps compared to the diffusion model, which requires many more steps. These improvements in inference time are clearly demonstrated in Table 2.
> >
> > ### Intention of Figure 2
> > The intention of Figure 2 was to compare the effects of consistency distillation (CD) and diffusion (Diff) models across different numbers of denoising steps. This comparison was meant to validate that CD consistently outperformed the diffusion model and that this effect was not coincidental.
> >
> > Regarding the use of five denoising steps for comparison, we focused on steps fewer than five because this is where CD demonstrates its advantage in inference time. By design, diffusion models achieve their best quality at higher denoising steps (10 in our case), but CD’s ability to achieve similar or better quality with fewer steps highlights its efficiency.
> >
> > ### Typographical Corrections
> > We thank the reviewer for carefully identifying typographical errors. We will address all the listed typos in the next revision and ensure the manuscript undergoes additional proofreading to avoid such issues.
> >
> > [1] Mariani, Giorgio, et al. "Multi-source diffusion models for simultaneous music generation and separation." in Proc. ICLR (2024).
> >
> > [2] Alexandre D  ́efossez, Nicolas Usunier, L  ́eon Bottou, and Francis Bach. Music source separation in
> > the waveform domain. arXiv preprint arXiv:1911.13254, 2019.

---

> > > ### Comment · Reviewer_Tp1m · 2024-11-27
> > >
> > > I thank the authors for their efforts in preparing the rebuttal. The explanations provided were helpful in understanding the proposed method in greater depth. However, I remain unconvinced, primarily due to the lack of sufficiently convincing validation. While I find the combination of deterministic models and consistency models an interesting and valuable approach, the experimental results do not sufficiently demonstrate that the proposed method can generalize to real-world data, even with sufficient training data. The claim that "diffusion models are data-hungry" is not, in my view, a sufficient justification for the observed limitations.
> > >
> > > It is worth noting that with access to abundant quantitative and qualitative data, many methodologies could be made to work effectively and yield strong results. However, in real-world scenarios, acquiring such data is often highly challenging. For research purposes, it is crucial to develop methods that remain effective even with limited data availability. I encourage the authors to further refine their work and enhance their approach to address these critical issues, and I truly look forward to seeing their progress in the future.

---

### Official Review · Reviewer_LTkM · 2024-11-04

**Soundness:** 3
**Presentation:** 3
**Contribution:** 3
**Rating:** 6
**Confidence:** 4

**Summary:**

This paper applies consistency distillation of diffusion models to the task of music source separation. The performance is evaluated on a standard task involving the separation of bass, drums, guitar, and piano. The proposed method is efficient, and the separation performance is better than conventional methods in terms of SISDR.

**Strengths:**

- The paper makes a valuable contribution by being one of the first to apply the latest technique, consistency distillation, to music source separation. While it may not exhibit groundbreaking originality, the work achieves a reasonable level of novelty within the field.
- The literature survey is well-executed, providing a thorough and balanced overview without notable omissions or excesses. The paper provides clear and comprehensive background information, along with a well-explained presentation of the methods used for comparison, making it easy for readers to understand the context.
- The proposed method is highly efficient. While there may be some concerns regarding the evaluation approach, it does show improvements over conventional methods in terms of the objective metric (SISDR).

**Weaknesses:**

- The task addressed is fairly standard, and similar studies on this task are quite common. In that sense, the work does not offer groundbreaking novelty. It would also be interesting to explore whether the method is effective for other types of music or instruments, expanding the scope of its applicability.
- A drawback of the proposed method is that it does not demonstrate particular effectiveness in out-of-distribution (OOD) domains, as highlighted in Appendix C. As the authors acknowledge, this limitation remains an open research question for future work.

**Questions:**

- To me, SISDR values presented in Table 1 did not seem to align well with the audio quality I perceived when listening to the samples provided on the linked website. When SISDR exceeds 20 dB, it is questionable whether this metric is suitable for evaluating the quality of source separation. Instead of relying solely on waveform-based metrics like SISDR, which are grounded in squared error, it would be worthwhile to consider alternative evaluation methods. For instance, metrics that emulate human auditory perception through neural networks might provide more meaningful insights.
- The detailed comparison of parameters in Figure 2 is commendable and adds valuable insight. However, I am curious whether $\sigma_{\max} = 0.01244, 0.2497, 0.2495$ were specifically chosen to optimize the performance shown in these graphs. It is important to clarify whether the selection of these values involved any potential use of the evaluation data for tuning purposes.

---

> ### Author Response · Authors · 2024-11-21
>
> ## Response to Strengths
> We thank the reviewer for highlighting the strengths of our paper. We appreciate your recognition of the novelty and efficiency of our proposed method, as well as the clarity of our literature survey and methodology presentation.
> We are glad the application of consistency distillation (CD) to music source separation is seen as a valuable contribution. While this is one important aspect of our work, we would like to emphasize that our other very important novelty lies elsewhere. Beyond introducing CD to source separation, we present a novel approach by merging discriminative and generative models for source separation, which we believe represents a significant contribution to the field.
> Thank you for your positive feedback on the literature survey and background information. We made a concerted effort to provide a balanced and comprehensive review, aiming to present related work through the lens of our proposed method, particularly its merging of discriminative and generative techniques.
> Finally, we appreciate your recognition of our method's efficiency and its improvements in the objective metric (SISDR). We believe the improvements shown in our results and the example files demonstrate the substantial impact of our approach.
>
> ## Response to Weaknesses
>
> ### Task Novelty
> We acknowledge that music source separation is a well-studied task. Our primary focus was to introduce consistency distillation and demonstrate its feasibility and benefits in this domain. Additionally, we proposed a novel combination of discriminative and generative approaches, which we believe contributes meaningfully to advancing methodology within this field. While our work may not redefine the task itself, we hope it paves the way for further innovation.
>
> ### Applicability to Other Music or Instruments
> We appreciate the suggestion to explore the effectiveness of our method on other types of music or instruments and see this as a potential continuation of our study. However, in this particular work, our primary focus was to test the proposed methods using a dataset that allows direct comparison with state-of-the-art baselines in the field.
>
> ### Out-of-Distribution (OOD) Effectiveness
> We agree that the proposed method’s performance in out-of-distribution domains, specifically on the MUSDB dataset, is a limitation. As acknowledged in the paper, this remains an open research question. We have proposed a possible explanation for this limitation, which we believe is related to the relatively small size of the MUSDB dataset. Addressing this issue is a priority for our future work.
>
> ## Response to Questions
>
> ### SISDR and Audio Quality Perception
> Thank you for the feedback. We believe the perceptual differences between the deterministic model and the diffusion and CD models are evident on the demo page and align well with the rise in SISDR values. We understand the concern regarding the deterministic model achieving SISDR >20 dB despite noticeable leakage and poor audio quality. However, we assure the reviewer that we thoroughly tested our code and evaluation methods, ensuring they are consistent with those used in baseline studies. We are confident that the reported SISDR values are accurate.
>
> While we acknowledge that SISDR may not fully capture human perception, it remains a widely accepted quantitative metric in the field. SISDR is particularly robust and reliable in source separation because it directly measures the difference between the generated and original sources. It provides a clear, quantifiable way to compare models and ensure progress. Additionally, as SISDR is a standard metric used by our baseline studies, it ensures comparability across methods.
>
> ### Selection of σmax Values
> Thank you for the question regarding the choice of σmax values. Our dataset was divided into training, validation, and test sets. The values (σmax=0.01244, 0.2497, 0.2495) were chosen through parameter search inference experiments on the validation data and were not specifically tuned using the test data. We will clarify this point in the manuscript to avoid any potential misunderstanding.

---

> > ### Comment · Reviewer_LTkM · 2024-11-26
> > **Comment by Reviewer**
> >
> > Thank you for your response. I am reasonably satisfied with your selection of $\sigma$-s.
> >
> > That said, regarding the first question, the evaluation methodology is somewhat weak and could benefit from further refinement, as other reviewers have also pointed out. Beyond merely competing within the constraints of existing benchmarks, it is important to push the frontiers of the field and explore new possibilities. Adopting this broader perspective could significantly enhance the impact of the research.

---

### Meta-Review · Area_Chair_svb1 · 2024-12-21

**Metareview:**

**Paper Summary:**

This paper applies consistency modeling (Song et. al., ICML 2023) to music source separation. The method is evaluated on the Slakh2100 (synthetic data) and MUSDB (real music) datasets.

**Strengths:**

The application of consistency models to source separation is interesting, as are the empirical performance improvements on Slackh2100. Multiple reviewers (LTkM, 2aY5) appreciated the quality and clarity of writing.

**Weaknesses:**

The experimental results are somewhat weak, as noted by all the reviewers. Superior performance is demonstrated on Slakh2100. On the more realistic MUSDB dataset, results are more mixed: outperforming several baselines, but underperforming more competitive methods (Mariani et al., ICLR 2024). Moreover, unlike for Slakh2100, "we did not observe any improvement from the diffusion and CD methods over the deterministic model when further trained on the MUSDB dataset."

**Conclusion**

I think this paper is not ready for publication. While the results on Slack2100 are intriguing, we should be cautious when drawing conclusions about synthetic data. I acknowledge that authors advance a plausible explanation for the weaker MUSDB results: namely, limited dataset size. But this is a challenge to be overcome when adapting generative modeling methods to data-poor domains like music source separation. I encourage the authors to think creatively and ambitiously about how to address this challenge.

**Additional Comments On Reviewer Discussion:**

Reviewers were ultimately not swayed by the author responses on the question of experiments and evaluation.

---

### Decision · Program_Chairs · 2025-01-22

Reject